# An Oral Rinse Active Matrix Metalloproteinase-8 Point-of-Care Immunotest May Be Less Accurate in Patients with Crohn’s Disease

**DOI:** 10.3390/biom10030395

**Published:** 2020-03-04

**Authors:** Jaana Rautava, Ulvi K. Gürsoy, Adrian Kullström, Eija Könönen, Timo Sorsa, Taina Tervahartiala, Mervi Gürsoy

**Affiliations:** 1Department of Oral Pathology and Oral Radiology, Institute of Dentistry, University of Turku, 20520 Turku, Finland; adrian.a.kullstrom@utu.fi; 2Department of Pathology, Turku University Hospital, 20521 Turku, Finland; 3Department of Periodontology, Institute of Dentistry, University of Turku, 20520 Turku, Finland; ulvi.gursoy@utu.fi (U.K.G.); eija.kononen@utu.fi (E.K.); mervi.gursoy@utu.fi (M.G.); 4Oral Health Care, Welfare Division, City of Turku, 20101 Turku, Finland; 5Department of Oral and Maxillofacial Disease, Helsinki University Hospital, University of Helsinki, 00014 Helsinki, Finland; timo.sorsa@helsinki.fi (T.S.); taina.tervahartiala@helsinki.fi (T.T.); 6Department of Dental Medicine, Karolinska Institute, 14152 Huddinge, Sweden

**Keywords:** collagenases, Crohn’s disease, dental caries, mouth, periodontitis

## Abstract

The diagnostic accuracy of point-of-care (PoC) applications may be compromised in individuals with additional inflammatory conditions. This cross-sectional study examined the performance of a commercial oral rinse active matrix metalloproteinase-8 (aMMP-8) PoC immunotest in individuals with (*n* = 47) and without Crohn’s disease (CD) (*n* = 41). Oral rinse collected from the participants was analyzed by the PoC immunotest. Molecular forms and fragments of salivary MMP-8 were detected by western immunoblotting. The sensitivity of the immunotest for periodontitis was 60.0% in the CD group and 90.0% in the control group. The respective specificity was 75.0% and 80.0%. In both groups, clinical diagnosis of periodontitis exhibited a significant association with the immunotest results, however, the odds ratio (OR) was more than ten-fold in controls (OR 54.3, 95% CI: 3.1–953, *p* = 0.006) in comparison to CD patients (OR 5.2, 95% CI: 1.3–21.6, *p* = 0.022). According to Western immunoblot results, the immunotest MMP-8 positivity was not related to elevated levels of molecular forms and fragments of MMP-8 in the CD group, as in the control group. The diagnostic accuracy of the aMMP-8 PoC oral rinse immunotest is reduced in CD patients, which may be related to lower levels or undetectable complexes.

## 1. Introduction

Crohn’s disease (CD) is a chronic inflammatory disease of the gastrointestinal tract, involving a complex interplay between genetic risk, environmental exposures, the microbiota, and the immune system. Individuals suffering from CD seem to be prone to developing periodontitis, caries, and oral mucosal lesions [1,2,3]. CD patients have a higher decayed-missed-filled teeth (DMFT) index than healthy individuals, which could be due to differences in their diet [3]. Indeed, CD patients reportedly exhibit an increased sugar consumption and have insufficient oral hygiene [4]. The suggested link between CD and periodontitis may be explained by shared pathways in their pathogenesis related to altered gut/oral microbiota [5,6] and immune responsiveness [7]. Both CD and periodontitis have been linked to a disrupted immunological response to the local microbiota. The innate immune system, particularly neutrophils and their inadequate function, plays a major role in the pathogenesis of these diseases [8,9,10]. Of tissue-degrading matrix metalloproteinases (MMPs), MMP-8 is the key inflammatory mediator in their inter-relationship; elevated MMP-8 levels have been detected in CD patients both in the intestine and in the oral cavity [11,12].

Periodontitis is diagnosed based on clinical signs of gingival bleeding, loss of clinical attachment, deepened periodontal pockets, and the presence of radiographical alveolar bone loss [13,14]. Yet, the use of these parameters carries limitations, since they do not recognize the onset of inflammation or identify individuals who are at risk of disease initiation or progression in a given population [15]. Instead, biomarker-based diagnostic methods are less laborious, non-invasive, and do not require trained dentists, being especially useful in studies with large study populations [16].

In periodontology, point-of-care (PoC) tests aim to detect periodontal diseases by analyzing the levels of biomarkers in oral fluids (saliva, oral rinse, and/or gingival crevicular fluid) without need of a laboratory [17,18]. Among the tested biomarkers, salivary and oral rinse levels of MMP-8 are shown to be effective in distinguishing periodontitis patients from periodontally healthy individuals [19,20,21]. For instance, PoC tests using active MMP (aMMP)-8 as a diagnostic marker with a threshold of 20 ng/mL can successfully identify periodontitis with 64–95% sensitivity and 60–100% specificity [22,23,24,25]. Notably, the test sensitivity improves in line with disease severity [22,24].

To date, no data exist on the performance of the selected oral rinse aMMP-8 PoC immunotest in individuals with compromised immune responsiveness. In the present case-control study, we hypothesized that CD, a systemic condition with impaired immune function, leads to decreased sensitivity and specificity of the PoC immunotest. Our aim was to evaluate the utility of the immunotest for chair-side diagnostics of periodontal disease in subjects with and without CD. As a main conclusion, we state that the accuracy of the immunotest was impaired in individuals with CD.

## 2. Materials and Methods

### 2.1. Study Design and Enrolment of Participants

This was a case-control study conducted at the Institute of Dentistry, University of Turku, from January 2017 to May 2018. The CD group, consisting of 47 CD patients between the ages of 22 and 77 years, was recruited from a Crohn and Colitis patient organization (IBD Association of Finland). Their diagnosis of CD had previously been confirmed by endoscopy and histopathological analysis of intestinal biopsy specimens. The control group, including 41 age- and gender-matched individuals without CD or other immune-related diseases, was recruited using advertisements placed at the Institute of Dentistry, University of Turku. Exclusion criteria for both groups included diabetes mellitus, smoking, excessive use of alcohol, less than 24 teeth, and the use of antimicrobial medication in the preceding six months prior to the clinical examination. In addition, individuals with periodontitis as a manifestation of systemic disease [26] as well as pregnant or lactating women were excluded.

The research protocol was approved by the Ethical Committee of the Hospital District of Southwest Finland (114/1801/2016), and the study was conducted according to the guidelines of the World Medical Association Declaration of Helsinki. Each participant was informed of the purpose, potential side effects and benefits of the study and gave their written consent. A structured questionnaire was collected from all participants regarding their demographic data (such as age, sex, and residence), general health, medications, and previous dental care. Additionally, the short Crohn’s disease activity index (CDAI) was used to assess the disease activity of the CD patients [27].

### 2.2. Clinical Oral Examination

The full-mouth clinical examination included registrations of oral mucosal status performed by a specialist in oral pathology (JR) together with cariological and periodontal measurements generated by a periodontist (MG). Before this study, the intra-examiner (MG) agreement was tested in double measurements of 10 patients and discovered to be very good.

The oral mucosal examination included visual inspection of the lips, mucous membranes of the oral cavity, and oropharynx while pressing the tongue down as well as manual palpation of the neck. All mucosal findings were described by clinical appearance and the site of the lesion. CD-associated mucosal lesions (sulcular ulceration, aphthous stomatitis, mucosal cobblestone appearance, lip swelling, angular cheilitis, mucogingivitis) were registered separately [28,29].

Cariological status was assessed based on visual inspection, fiber-optic transillumination and tactile examination on the five surfaces of each tooth. Additionally, initial caries lesions together with other hard tissue defects on tooth surfaces, such as erosion and attrition, were determined. Caries prevalence in the whole dentition (excluding the third molars) was expressed using the decayed-missed-filled surface (DMFS) index. The participants were classified as having caries when they had at least one tooth surface with a carious lesion.

Periodontal status was obtained from six sites per each natural tooth and dental implant using a WHO probe (LM-Instruments Oy, Parainen, Finland) with a ball-tip end diameter of 0.5 mm. A full-mouth dichotomous visible plaque index (VPI) score was recorded as percentage. Gingival recession (REC) was detected as the distance between the cemento-enamel junction and free gingival margin, while probing pocket depth (PPD) was measured from the free gingival margin to the most apical penetration point of the probe. Loss of clinical attachment level (CAL) was determined as the sum of REC and PPD. A full-mouth dichotomous gingival bleeding score was obtained upon bleeding on probing (BOP) and presented as percentage. In addition, furcation defects and tooth mobility scores were recorded.

Individuals with BOP < 15%, PPD < 4 mm, and no loss of CAL were regarded as periodontally healthy, whereas gingivitis was diagnosed in the presence of BOP ≥15% without periodontal pockets and loss of CAL [30]. The diagnostic criteria for periodontitis included PPD ≥ 4 mm and/or loss of CAL on ≥ 2 non-adjacent teeth together with BOP [31]. The severity of the disease was classified based on PPD and loss of CAL as follows: initial periodontitis (PPD 4 mm, CAL loss 1–2 mm), moderate periodontitis (PPD 5–6 mm, CAL loss 3–4 mm), and severe periodontitis (PPD ≥ 7 mm, CAL loss ≥ 5 mm). After the clinical examination, the patient was informed both orally and in written form of the findings. The patients with need for treatment were advised to contact their own dentist.

### 2.3. Application of an Oral Rinse aMMP-8 PoC Immunotest

The study participants were instructed to avoid tooth brushing, eating, and chewing gum 30 min prior to the visit. Oral rinse sampling was performed before the clinical oral examination. First, each participant pre-rinsed the mouth with tap water for 30 s and then spitted out or swallowed the fluid. After 60 s of waiting, they re-rinsed vigorously the mouth with 5 mL of rinsing fluid^‡^ for 30 s and then poured the entire sample into a measuring cup, from which the sample was drawn up into a syringe and used for the aMMP-8 PoC analysis.

The immunotest (PerioSafe^®^ PRO, Dentognostics GmbH, Jena, Germany) was used according to the manufacturer’s instructions. A single examiner (UKG) ran all tests, and the result was interpreted by visual estimation 5 min after testing. The outcome was defined as positive (+) when a double line (i.e., both test and control line) was visible on the test window or negative (−) when only a single line was observed [22].

### 2.4. MMP-8 Western Immunoblot

Stimulated saliva samples were collected for 5 min from the patients, separately from the immunotest. A modified enhanced chemiluminescence (ECL) western blotting kit (GE Healthcare, Amersham, UK) was used to detect the molecular forms of MMP-8 from the stimulated saliva as described earlier [32,33]. Salivary flow levels were measured to minimize the variations in sample collection and sample loading during immunoblottings. Briefly, the saliva samples were mixed with Laemmli’s buffer without any reducing reagents. The samples were heated for 5 min, followed by protein separation with 11% sodium dodecyl sulphate (SDS)-polyacrylamide gels. The proteins were then electro-transferred onto nitrocellulose membranes (Protran, Whatman GmbH, Dassel, Germany). Milk powder (5%; Valio Ltd., Helsinki, Finland) was used to block non-specific binding in TBST buffer (10 mM Tris-HCl, pH 7.5, containing 22 mM NaCl and 0.05% Triton-X) for 1 h. The membranes were incubated with polyclonal primary antibodies anti-MMP-8 [34] overnight and followed by the horseradish peroxidase-linked secondary antibody (GE Healthcare, Buckinghamshire, UK) for 1 h. The target proteins were visualized using the ECL system, and scanned and converted into arbitrary units (relative levels of mean intensity) using GS-700 Imaging Densitometer Scanner (BioRad, Hercules, CA, USA) and Bio-Rad Quantity One program. Purified human activated MMP-8 (BioTeZ, Berlin, Germany) was used as positive control and low range prestained SDS-PAGE Standards (BioRad) served as a molecular weight marker.

### 2.5. Statistical Analyses

All statistical analyses were performed using the SPSS statistical program (version 24.0; IBM Inc., Armonk, NY, USA). The Chi square test was used for comparing demographic and clinical data (with the exception of age and number of teeth) between the CD and control groups. Age, salivary flow rate, and number of teeth were compared between the groups with one-way ANOVA test. Multinomial regression analyses were performed to describe the associations between periodontal diseases and the immunotest result, after adjusting for furcation and attrition defects. Statistical significance was defined as *p* < 0.05.

## 3. Results

### 3.1. Crohn’s Disease Status

Of the 47 CD patients, 89.4% received medication for CD and 34.0% had undergone surgical treatment for CD. According to the short CDAI score, 53.2% of the subjects were in remission (short CDAI < 150 points), 31.9% had mild disease (short CDAI 150–219 points), 14.9% had moderate disease (short CDAI 220–450 points), and none had severe disease (CDAI > 450 points).

### 3.2. Oral Health Status

Characterization of the CD and control groups on the basis of periodontal findings is presented in Table 1.

The DMFS index was 35.8 for the CD patients and 23.9 for the controls (*p* = 0.015). At least one dentin caries lesion was found in 31.9% of the CD patients and 12.2% of the controls (*p* = 0.028). The mean stimulated salivary flow was 1.42 mL/min and 1.59 mL/min, respectively. CD-associated mucosal lesions were detected in 10.6% of the CD patients. Mucosal lesions not associated with CD were found in 34.0% of the CD patients and 36.6% of the controls.

### 3.3. Oral Rinse aMMP-8 PoC Immunotest

The outcomes of the aMMP-8 PoC immunotest were tested against various clinical outcomes and diagnoses in terms of its sensitivity and specificity as presented in Table 2. Sensitivity and specificity values of the aMMP-8 PoC immunotest for detecting periodontitis were 60.0% and 75.0% in the CD patients, 90.9% and 80.0% in the controls, and 73.1% and 77.4% in the whole population, respectively.

The results of the aMMP-8 PoC immunotest were significantly associated with the diagnosis of periodontitis in the CD patients (OR: 5.2, 95% CI: 1.26–21.6, *p* = 0.022), in the controls (OR: 54.3, 95% CI: 3.09–953, *p* = 0.006), and in the whole population (OR: 8.5, 95% CI: 2.88–25.3, *p* < 0.001), as shown in Table 3.

### 3.4. MMP-8 Immunoblot

Figure 1 presents the relative levels of molecular forms, activations, and fragments of MMP-8 in saliva.

According to the immunoblot results, in the control group the PoC immunotest positivity is related to elevated levels of different forms and fragments of MMP-8 in saliva (Table 4). In CD patients, however, no difference in the levels of salivary forms of MMP-8 was observed between the PoC immunotest positive and negative individuals.

## 4. Discussion

To our knowledge, this is the first study to investigate the sensitivity and specificity of a chairside diagnostic kit in detection of periodontitis in patients with disrupted immune responsiveness. Our findings demonstrate that both the sensitivity and specificity of the aMMP-8 oral rinse PoC immunotest are significantly impaired in CD patients.

A comprehensive examination of periodontal, cariological, and oral mucosal health is the strength of the present study. Although there was no significant decrease in salivary flow in the CD group, the DMFS index was significantly higher in CD patients than in controls. Only 25% of the CD patients were periodontally healthy compared to 37% of the control group. In addition, CD patients presented more gingivitis and initial periodontitis than their generally healthy controls. These clinical findings are in line with several previous studies indicating differences in caries and periodontal states of CD patients in comparison to systemically healthy individuals [1,2,3]. On one hand, a higher DMF index may be attributed to nutritional and behavioral changes, such as an increased intake of carbohydrates and alterations in bacterial conditions in the oral cavity favoring mutans streptococci and lactobacilli [4,34,35], although these factors were not examined in the present study. On the other hand, the association between CD and periodontitis has been attributed to common predisposing factors, including age, genetic predisposition, and lifestyle factors, but perhaps more importantly to defects in the mucosal barrier, disrupted immune responses, and the exuberant host-response to bacteria leading to chronic inflammation [36]. Moreover, CD patients seem to exhibit a higher prevalence and more generalized and more severe periodontitis than non-CD individuals [2,3,37,38]. Contrarily, in the present study, severe periodontitis (12%) was diagnosed only in the control group. The small group size may have an impact on the results, but it was not possible to expand the present study sample size due to the low prevalence of Crohn’s disease. The cross-sectional study design is another limitation of the present study, and thus, we cannot infer any temporal association between CD and oral disease. Moreover, the recruitment of the control group was in part undertaken at a dental clinic, which may result in a selection bias. However, this does not weaken the associations between the outcomes of the chairside test and periodontitis diagnosis after adjusting for clinical parameters.

The PoC oral rinse and salivary tests can be used as noninvasive and economic methods for screening and diagnosing disease. Various inflammatory proteins present in saliva have been tested as candidate biomarkers for periodontitis [39,40,41,42,43]. For example, salivary MMP-8 can be detected in different isoforms based on its origin (neutrophilic or mesenchymal type) or molecular forms (active, pro-, fragmented, complex) [32,33]. Moreover, neutrophils, fibroblasts, and endothelial cells are the main sources of MMP-8 in the oral cavity [44,45]. Some studies have indicated that dental caries has an impact on oral MMP-8 levels, and thereby, caries lesions may affect the outcome of oral rinse aMMP-8 PoC immunotests [22,46]. The causal relationship between caries and salivary MMP-8 has not been established, but the association between the number of cariotic lesions and MMP-8 levels suggests a role for the collagenase in the dentin caries process [46]. In our study, the sensitivity and specificity of the kit for caries detection in the whole population were low, 55.0% and 67.6%, respectively. One explanation for this could be the activity of the cariotic process. In comparison to chronic or stable caries lesions, active lesions seem to correlate more with MMP activity [47]. In addition, open caries lesions may more likely cause leakage of MMPs into the saliva in comparison to non-cavitated small caries lesions [47].

Recent research has shown promising results regarding the use of aMMP-8 as a diagnostic biomarker in PoC oral rinse tests for periodontitis [22,23,24,25,43,48]. The oral rinse aMMP-8 PoC immunotest, also used in the present study, applies lateral flow chromatography and detects the active form of MMP-8 with targeted antibodies [43,48]. A positive result of the test is based on a cutoff of 20 ng/mL for aMMP-8, and it can identify periodontitis with 64–95% sensitivity and 60–100% specificity [22,23,24,25]. In comparison to salivary MMP-8 analyses, the oral rinse aMMP-8 PoC immunotest has demonstrated a higher accuracy when differentiating periodontal health from the disease [43,48]. However, little information is still available regarding the limitations of this test.

There are common and discordant elements in the pathogenesis of CD and periodontitis. Indeed, both disorders lead to a substantial defect in the mucosal barrier and exuberant host response to bacteria, resulting in inflammation and tissue destruction. It is reasonable to speculate that the dysregulation of the immune system in CD may have an effect on MMP-8 levels in the oral cavity. It has been shown in a mouse model simulating inflammatory bowel disease that inflammation in the colon changes the microbiota not only in the colon but also in the oral cavity [9]. In the current study, the sensitivity and specificity estimates of the aMMP-8 PoC immunotest for detecting periodontitis were found to be poorer in CD patients (60.0% and 75.0%) than in systemically healthy controls (90.9% and 80.0%). The reasons for this are most likely multifactorial. Firstly, it can be argued that the prescribed immunomodulatory or anti-inflammatory medications affect the test accuracy. However, these medications act as suppressors of inflammatory response, which means that both clinical and sub-clinical indices of periodontitis (i.e., bleeding on probing as well as MMP-8 expression and activation, respectively) can be restrained. Thus, the use of medications does not fully explain the low sensitivity and specificity values of aMMP-8 PoC test in periodontitis patients with Crohn’s disease. Secondly, impaired neutrophil MMP-8 function might be another explanation for the low sensitivity of the test in CD patients. Our western immunoblot experiment of salivary MMP-8 indicated that the aMMP-8 PoC oral rinse immunotest positivity in the CD group does not reflect the changes in the levels of molecular forms and fragments of MMP-8, varying from that observed in the control group. The difference may be explained, at least in part, by genetics. Indeed, a mutation in the nucleotide-binding oligomerization domain 2/caspase recruitment domain 15 (NOD2/CARD15) gene, which impairs the ability to recognize bacterial components and thus triggers an inadequate immune response, has been reported in patients with CD [49]. Interestingly, when patients suffering from rheumatoid arthritis - another example of a disease with dysregulated immune system - were compared to systemically healthy periodontitis patients, they exhibited increased aMMP-8 levels in gingival crevicular fluid [50]. Additionally, in presence of vascular leakage components in saliva samples, alpha 2-macroglobulin may form complexes with proteases, which in turn may partly explain the high-molecular weight complexes [51]. Formation of these complexes, however, may not totally explain the current results, since all samples were free of blood contamination. Finally, all these observations indicate that patients with a systemic disease manifested with impaired immune responsiveness need to be evaluated in a disease-specific manner before any diagnostic test can be reliably utilized. The reasons why (1) MMP-8 is not a functioning biomarker in CD patients or (2) aMMP-8 PoC immunotest seem to produce less accurate results in CD patients remain unsolved. Therefore, further studies are still required.

## 5. Conclusions

In conclusion, the performance of the oral rinse aMMP-8 PoC immunotest for recognizing periodontitis seem to be less reliable in CD patients than in systemically healthy individuals.

## 6. Patents

Professor Timo Sorsa is an inventor of US patents 5652223, 5736341, 5866932, 6143476, 20145192, 15/121801 and US 10,488,415 B2.

## Figures and Tables

**Figure 1 biomolecules-10-00395-f001:**
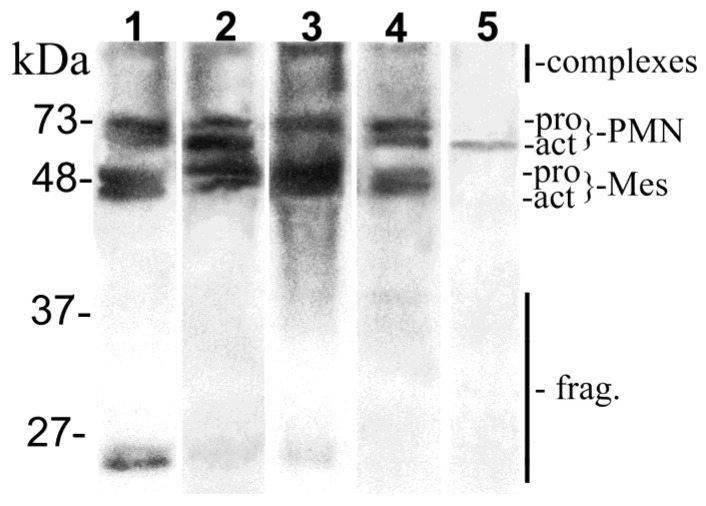
Western immunoblot analysis of salivary samples studied for molecular forms, degree of activation and related fragmentation of matrix metalloproteinase (MMP)-8. “PMN” indicates neutrophil-type collagenase (70–80 kDa) and “Mes” fibroblast-type MMP-8 collagenase (50–65 kDa) together with their pro and active forms. “Complexes” indicate multiple forms of MMP-8 (>110 kDa) and “fragments” the fragmented MMP-8 species (20–30 kDa). Lanes 1 and 2 illustrate the Crohn’s disease group, and lanes 3 and 4 the control group. All laboratory conditions (incl. time of exposure) and image analyses were standardized. Purified human activated MMP-8^##^ was used as positive control (lane 5). Mobilities of the molecular weight markers (low range prestained SDS-PADE Standards (Bio-Rad Laboratories, Inc, Richmond, CA, USA) are indicated on the left.

**Table 1 biomolecules-10-00395-t001:** Characterization of the study groups (patients with Crohn’s disease and their age- and gender-matched healthy controls) according to periodontal health status and stimulated salivary flow. PPD = probing pocket depth; BOP = bleeding on probing.

Findings	Crohn’s Disease Group (N = 47)	Control Group (N = 41)	Difference *P*-Value
Gender (male %)	23.4	29.3	0.629
Age, years (mean, st.dev)	46.4 (13.9)	47.4 (13.2)	0.734
No. of teeth (mean, st.dev)	27 (2.3)	27.6 (2.4)	0.192
VPI % (mean, st.dev)	23.8 (20.8)	22.9 (18.7)	0.828
BOP % (mean, st.dev)	32.7 (23.3)	33.9 (28.6)	0.820
Individuals with PPD 4 mm (n, %)	22 (46.8)	13 (31.7)	0.192
Individuals with PPD 5–6 mm (n, %)	1 (2.1)	8 (19.5)	**0.011**
Individuals with PPD ≥ 7 mm (n, %)	0	5 (12.2)	**0.019**
Individuals with REC ≥ 1 mm (n, %)	33 (70.2)	28 (68.3)	1.000
Individuals with REC ≥ 3 mm (n, %)	14 (29.8)	12 (29.3)	0.073
Periodontal diagnosis			**0.021**
Periodontally healthy (n, %)	12 (25)	15 (36.6)	
Gingivitis (n, %)	20 (42.6)	15 (36.6)	
Initial periodontitis (n, %)	14 (29.8)	4 (9.8)	
Moderate periodontitis (n, %)	1 (2.1)	2 (4.9)	
Severe periodontitis (n, %)	0 (0)	5 (12.2)	
Stimulated salivary flow, ml/min (mean, st.dev)	1.42 (0.72)	1.59 (0.90)	0.313

**Table 2 biomolecules-10-00395-t002:** Sensitivity and specificity of the oral rinse active matrix metalloproteinase-8 oral rinse point-of-care immunotest in relation to the presence of probing pocket depth (PPD), bleeding on probing (BOP), furcation defects, attrition, periodontal diseases (gingivitis or periodontitis), and periodontitis. (- = no individuals with PPD ≥ 7 mm)

Periodontal Findings	Sensitivity/Spesificity	Crohn’s Disease Group	Control Group	Whole Population
		(N = 47)	(N = 41)	(N = 88)
PPD 4 mm	Sensitivity	54.5	84.6	65.7
	Specificity	80	82.1	81.1
PPD 5–6 mm	Sensitivity	100	87.5	88.9
	Specificity	65.2	72.7	68.4
PPD ≥ 7 mm	Sensitivity	-	100	100
	Specificity	-	69.4	66.3
BOP ≥ 15%	Sensitivity	37.1	57.7	45.9
	Specificity	66.7	93.3	81.5
Furcation defect	Sensitivity	40	66.7	54.5
	Specificity	64.9	72.4	68.2
Attrition	Sensitivity	52.4	38.9	46.2
	Specificity	76.9	60.9	69.4
Periodontal diseases	Sensitivity	35.3	57.7	45
	Specificity	61.5	93.3	78.6
Periodontitis	Sensitivity	60	90.9	73.1
	Specificity	75	80	77.4

**Table 3 biomolecules-10-00395-t003:** Unadjusted and adjusted (furcation defects and attrition) associations of the selected oral rinse active matrix metalloproteinase 8 point-of-care immunotest with periodontal diseases (gingivitis or periodontitis) or with periodontitis. Data are given as odds ratio (95% confident intervals), and *p*-value.

Group	Periodontal Diseases		Periodontitis	
	Unadjusted	Adjusted	Unadjusted	Adjusted
Crohn’s disease group (N = 47)	0.9 (0.23–3.26), 0.84	1.3 (0.29–5.58), 0.752	4.5 (1.22–16.6), 0.024	5.2 (1.26–21.6), 0.022
Control group (N = 41)	19.1 (2.17–167), 0.08	20.4 (1.92–216), 0.012	40 (4.25–376), 0.001	54.3 (3.09–953), 0.006
Whole population (N = 88)	3.0 (1.06–8.45), 0.038	4.16 (1.34–12.9), 0.014	9.3 (3.25–26.6), <0.001	8.5 (2.88–25.3), <0.001

**Table 4 biomolecules-10-00395-t004:** Relative levels of mean intensity (median, min-max) of molecular forms and fragments of salivary MMP-8 as regards to oral rinse point-of care immunotest positivity (+) and negativity (−). *p* values indicating a significant difference (<0.05) are bolded.

Group		Complexes of MMP-8	PMN		Mesenchymal		Fragments of MMP-8	Total MMP-8
			Pro MMP-8	aMMP-8	Pro MMP-8	aMMP-8		
**Crohn’s disease Group**	**+ (N = 17)**	**1.8 (0.6–4.1)**	**0.4 (0.1–2.1)**	0.0 (0.0–1.2)	0.6 (0.0–1.5)	0.5 (0.0–1.6)	0.1 (0.0–1.4)	3.8 (1.0–9.5)
	**− (N = 30)**	1.5 (0.2–4.2)	0.4 (0.0–1.1)	0.1 (0.0–1.3)	0.8 (0.0–1.6)	0.8 (0.0–1.7)	0.6 (0.0–0.5)	3.7 (1.5–8.2)
	***P***	0.330	0.877	0.708	0.303	0.061	0.192	0.982
**Control Group**	**+ (N = 16)**	1.4 (0.1–4.7)	0.6 (0.0–1.5)	0.4 (0.0–1.4)	1.0 (0.1–1.8)	0.9 (0.0–1.9)	0.1 (0.0–2.4)	4.9 (0.4–11.8)
	**− (N = 25)**	0.5 (0.0–3.7)	0.2 (0.0–1.8)	0.0 (0.0–1.0)	0.6 (0.0–1.6)	0.7 (0.0–1.7)	0.0 (0.0–0.5)	2.2 (0.2–7.6)
	***P***	**0.022**	**0.037**	**0.015**	0.121	0.209	**0.046**	**0.015**
**Whole** **Population**	**+ (N = 33)**	1.6 (0.1–4.7)	0.4 (0.0–2.1)	0.1 (0.0–1.4)	0.8 (0.0–1.8)	0.2 (0.0–1.9)	0.1 (0.0–2.4)	3.9 (0.4–11.8)
	**− (N = 55)**	1.0 (0.0–4.2)	0.3 (0.0–1.8)	0.5 (0.0–1.3)	0.7 (0.0–1.6)	0.7 (0.0–1.7)	0.1 (0.0–0.5)	3.1 (0.2–8.2)
	***P***	**0.032**	0.174	0.140	0.670	0.689	**0.020**	0.062

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
