# Peer review of "An Oral Rinse Active Matrix Metalloproteinase-8 Point-of-Care Immunotest May Be Less Accurate in Patients with Crohn’s Disease"

_biomolecules, 2020, doi:10.3390/biom10030395_

Round 1

Reviewer 1 Report

The authors have now addressed all my concerns of previous review rounds.

Reviewer 2 Report

This is an important and well written manuscript describing the outcomes of a clinical study investigating the accuracy of the deployment of a point of care diagnostic test for periodontitis  in patients with crohn's disease compared to non-crohn's disease controls. The rational and execution of the study is clear and limitations are discussed. Apprportiate analysis has been carried out and deeper investigation has been presented. An appropriate conclusion has been stated and a desire for further research to deepen the explanation of the phenomenon described. No changes to the manuscript are requested.

This manuscript is a resubmission of an earlier submission. The following is a list of the peer review reports and author responses from that submission.

Round 1

Reviewer 1 Report

In this manuscript, Rautava et al. evaluate the effect of Crohn's disease on the predictive value of on oral rinse active MMP-8 point-of-care test. The authors compare results from this test with Western-blot analysis of similar samples from the same individuals. The most interesting conclusion is that the test is less powerfull in individuals with Crohn's disease and that changes in the forms of MMP-8 can be observed in patients with Crohn's disease. I really appriciate this comparison and the fact that the authors critically analyse all MMP-8 forms, fragments and complexes. However, I still have some points which need further clarification:

abstract: the fist line of the abstract is broad. it states that PoC applications may be less accurate in individuals with disrupted immune response. This is a very general statement which requires more analysis of different tests to be justified. Please be more specific.

abstract: in the last line the authors write: "which may be related to their diminished MMP-8 response." This is a very broad statement that doesnt have much meaning. Please be more specific.

Regarding the western blot analysis: how were the values calculated. How was variation in sample loading, sample collection,... taken into account?

Please add western blot images showing the differences in samples and with indication of the fragments selected for visualization in table 4. It is always very usefull to see such images since it gives much more information than a table.  

Table 4 could be improved: What is the unit of the values shown in the table?  How did you distinguish between PMN and Mesenchyma MMP-8? (and why are these different?) please also indicate sample sizes in the table (n = ?)

How exactly does the PoC test work? is it based on the degradation of an MMP-8 substrate (hence the detection of active MMP-8). Or, is it based on a real immunoassay where an antibody is used to detect activated MMP-8? In addition, how specific is this test for MMP-8? This is very usefull information and helps to estimate eactly which forms the PoC test is detecting.

Is anything known about the presence of components derived from the blood in samples from patients with Crohn's disease? Any bleeding or vascular leakage would mean that inhibitors such as alpha-2-macroglobulin could react with any active MMP-8 and convert all active MMP-8 into a high-molecular weight complex detected by non-reducing PAGE (as used in this study) (as shown recently for MMP-9, see: PMID:31642940). This could explain some of the differences seen in the presence of high-molecular weight MMP-8.

Discussion: Line 230: "Resent" should be "Recent"

Author Response

We thank all the Reviewers for insightful comments on our manuscript. Please, find the point-by point answers to the comments below. The line numbers mentioned below are the one showing in the manuscript version where all correction marks are visible.

Reviewer 1:

Comment 1: abstract: the fist line of the abstract is broad. it states that PoC applications may be less accurate in individuals with disrupted immune response. This is a very general statement which requires more analysis of different tests to be justified. Please be more specific.

Answer 1: We have modified the first sentence: “The diagnostic accuracy of point-of-care (PoC) applications may be compromised in individuals with disrupted immune responsiveness”. The word limit does not allow a wider background in the abstract but we specify more in the introduction lines 277-283 and also lines 284-289.

Comment 2: abstract: in the last line the authors write: "which may be related to their diminished MMP-8 response." This is a very broad statement that doesnt have much meaning. Please be more specific.

Answer 2: Unfortunately, we are not able to open the issue in the abstract within word limits. However, we have made an effort to discuss the matter carefully in lines 908-1171. 

Comment 3: Regarding the western blot analysis: how were the values calculated. How was variation in sample loading, sample collection,... taken into account?

Answer 3: The western blot analysis protocol (sample collection, sample loading, staining, handling, densitometric computerized quantification) and the antibody are described in details in the following publications: Hanemaajier et al., J Biol Chem 1997;272:31504-9; Prikk et al., Lab Invest 2002;82:1535-45; Kiili et al., Clin Perio 2002;29:224-32; Buduneli et al., J Periodontol 2011;82:716-25; Gursoy et al., J Clin Period 2018;45:1421-1228. We have added references and text in lines 495-500.

Comment 4: Please add western blot images showing the differences in samples and with indication of the fragments selected for visualization in table 4. It is always very usefull to see such images since it gives much more information than a table.  

Answer 4: Representative western immunoblot image (Figure 1) and a figure legend has been constructed to the revised manuscript. 

Comment 5: Table 4 could be improved: What is the unit of the values shown in the table?  How did you distinguish between PMN and Mesenchyma MMP-8? (and why are these different?) please also indicate sample sizes in the table (n = ?)

Answer 5: The Table 4 has been improved in the revised manuscript. The sample sizes have been added. The unit of value is arbitrary units obtained and used as described in detail by Kiili et al., Clin Perio 2002;29:224-32; Buduneli et al., J Periodontol 2011;82:716-25; Gursoy et al., J Clin Period 2018;45:1421-1228. PMN and Mes/fibroblastic MMP-8 are differentiated by differences in their molecular sizes (Hanemaajier et al., J Biol Chem 1997;272:31504-9) by western blots and quantitated by computerized denstrometric scannings (the same refs as above). The references are now added in the materials and methods part of the revised manuscript.

Comment 6: How exactly does the PoC test work? is it based on the degradation of an MMP-8 substrate (hence the detection of active MMP-8). Or, is it based on a real immunoassay where an antibody is used to detect activated MMP-8? In addition, how specific is this test for MMP-8? This is very usefull information and helps to estimate eactly which forms the PoC test is detecting.

Answer 6: The chapter 2.3. in lines 482-492 has been improved and a reference has been added in the revised manuscript. More discussion on the matter has been added in lines 880-886. The information of PoC test is also found under the US patents 5652223, 5736341, 5866932, 6143476, 20145192 and 15/121801 which are also included in the manuscript as chapter 6.

Comment 7: Is anything known about the presence of components derived from the blood in samples from patients with Crohn's disease? Any bleeding or vascular leakage would mean that inhibitors such as alpha-2-macroglobulin could react with any active MMP-8 and convert all active MMP-8 into a high-molecular weight complex detected by non-reducing PAGE (as used in this study) (as shown recently for MMP-9, see: PMID:31642940). This could explain some of the differences seen in the presence of high-molecular weight MMP-8.

Answer 7: Our oral fluid samples were free of blood contamination. The saliva collection and PoC test were performed prior to clinical oral examination in order to avoid blood leakage. This has now been explained in the manuscript in lines 483-488. We have added text in the revised manuscript in lines 1173-1177 as following: “Additionally, in presence of vascular leakage components in saliva samples, alpha 2-macroglobulin may form complexes with proteases, which may partly explain the high-molecular weight complexes [51]. However, formation of these complexes may not totally explain our results, as our samples were free of blood contamination”.

Comment 8: Discussion: Line 230: "Resent" should be "Recent"

Answer 8: Sorry for the typo, it has been corrected.

Reviewer 2 Report

Thank you for all authors good work

The article deals with Crohn’s disease interferes with the diagnostic 2 accuracy of an oral rinse active matrix 3
 metalloproteinase-8 point-of-care immunotest

The topic is very important and urgent, and this study introduces a novel and promising approach. The research design is also appropriate.

However, the language of the manuscript is poor, there are few important mistakes, and especially the Introduction and MATERIALS AND METHODS section is unacceptable need rewriting and improve

Author Response

Reviewer 2.

We thank all the Reviewers for insightful comments on our manuscript. Please, find the point-by point answers to the comments below. The line numbers mentioned below are the one showing in the manuscript version where all correction marks are visible.

The article deals with Crohn’s disease interferes with the diagnostic 2 accuracy of an oral rinse active matrix 3
 metalloproteinase-8 point-of-care immunotest

The topic is very important and urgent, and this study introduces a novel and promising approach. The research design is also appropriate.

However, the language of the manuscript is poor, there are few important mistakes, and especially the Introduction and MATERIALS AND METHODS section is unacceptable need rewriting and improve

The materials and methods section has been re-written and the language has been revised by an English expert.

Reviewer 3 Report

1) The paper is in need of editing for grammatical irregularities, but also suffers from a lack of detail and clarity. The immunotest was designed to detect the active form on MMP-8, but no information is given as to if or how well it detects other forms, or what the limits of detection are.  The physiological significance of the different forms is not really explained either.  The legends for the tables should be more informative, and the N numbers for the groups should be given in each case.  Are the apparent differences in sensitivity and specificity given in Table 2 statistically different between Crohn’s and Control?  What do the numbers in Table 4 represent?  The mesenchymal active form in the whole population lists a median of 6.2 but a max of 1.9.  The formatting of Table 4 is messed up.

2) The study is small, especially when subdivided, and the incidence of periodontal disease in the groups is different (the patients with Crohn’s have more mild periodontal disease, with only 2% moderate and no severe disease, whereas the 17% of controls have moderate or severe disease).  The impact of this on results should be more adequately addressed.

3) An attempt is made to correlate levels of the various forms of MMP-8 in saliva of people who tested positive vs negative by immunotest by immunoblot.  No immunoblots are shown and although the methods section describes densitometic analysis, that isn’t in the legend and there is no information as to what the numbers in the table represent or how many samples are in each group.  Immunoblots are at best only semi-quantitative, so it unclear what significance this data can have.  

Author Response

We thank all the Reviewers for insightful comments on our manuscript. Please, find the point-by point answers to the comments below. The line numbers mentioned below are the one showing in the manuscript version where all correction marks are visible.

Reviewer 1:

Comment 1: abstract: the fist line of the abstract is broad. it states that PoC applications may be less accurate in individuals with disrupted immune response. This is a very general statement which requires more analysis of different tests to be justified. Please be more specific.

Answer 1: We have modified the first sentence: “The diagnostic accuracy of point-of-care (PoC) applications may be compromised in individuals with disrupted immune responsiveness”. The word limit does not allow a wider background in the abstract but we specify more in the introduction lines 277-283 and also lines 284-289.

Comment 2: abstract: in the last line the authors write: "which may be related to their diminished MMP-8 response." This is a very broad statement that doesnt have much meaning. Please be more specific.

Answer 2: Unfortunately, we are not able to open the issue in the abstract within word limits. However, we have made an effort to discuss the matter carefully in lines 908-1171. 

Comment 3: Regarding the western blot analysis: how were the values calculated. How was variation in sample loading, sample collection,... taken into account?

Answer 3: The western blot analysis protocol (sample collection, sample loading, staining, handling, densitometric computerized quantification) and the antibody are described in details in the following publications: Hanemaajier et al., J Biol Chem 1997;272:31504-9; Prikk et al., Lab Invest 2002;82:1535-45; Kiili et al., Clin Perio 2002;29:224-32; Buduneli et al., J Periodontol 2011;82:716-25; Gursoy et al., J Clin Period 2018;45:1421-1228. We have added references and text in lines 495-500.

Comment 4: Please add western blot images showing the differences in samples and with indication of the fragments selected for visualization in table 4. It is always very usefull to see such images since it gives much more information than a table.  

Answer 4: Representative western immunoblot image (Figure 1) and a figure legend has been constructed to the revised manuscript. 

Comment 5: Table 4 could be improved: What is the unit of the values shown in the table?  How did you distinguish between PMN and Mesenchyma MMP-8? (and why are these different?) please also indicate sample sizes in the table (n = ?)

Answer 5: The Table 4 has been improved in the revised manuscript. The sample sizes have been added. The unit of value is arbitrary units obtained and used as described in detail by Kiili et al., Clin Perio 2002;29:224-32; Buduneli et al., J Periodontol 2011;82:716-25; Gursoy et al., J Clin Period 2018;45:1421-1228. PMN and Mes/fibroblastic MMP-8 are differentiated by differences in their molecular sizes (Hanemaajier et al., J Biol Chem 1997;272:31504-9) by western blots and quantitated by computerized denstrometric scannings (the same refs as above). The references are now added in the materials and methods part of the revised manuscript.

Comment 6: How exactly does the PoC test work? is it based on the degradation of an MMP-8 substrate (hence the detection of active MMP-8). Or, is it based on a real immunoassay where an antibody is used to detect activated MMP-8? In addition, how specific is this test for MMP-8? This is very usefull information and helps to estimate eactly which forms the PoC test is detecting.

Answer 6: The chapter 2.3. in lines 482-492 has been improved and a reference has been added in the revised manuscript. More discussion on the matter has been added in lines 880-886. The information of PoC test is also found under the US patents 5652223, 5736341, 5866932, 6143476, 20145192 and 15/121801 which are also included in the manuscript as chapter 6.

Comment 7: Is anything known about the presence of components derived from the blood in samples from patients with Crohn's disease? Any bleeding or vascular leakage would mean that inhibitors such as alpha-2-macroglobulin could react with any active MMP-8 and convert all active MMP-8 into a high-molecular weight complex detected by non-reducing PAGE (as used in this study) (as shown recently for MMP-9, see: PMID:31642940). This could explain some of the differences seen in the presence of high-molecular weight MMP-8.

Answer 7: Our oral fluid samples were free of blood contamination. The saliva collection and PoC test were performed prior to clinical oral examination in order to avoid blood leakage. This has now been explained in the manuscript in lines 483-488. We have added text in the revised manuscript in lines 1173-1177 as following: “Additionally, in presence of vascular leakage components in saliva samples, alpha 2-macroglobulin may form complexes with proteases, which may partly explain the high-molecular weight complexes [51]. However, formation of these complexes may not totally explain our results, as our samples were free of blood contamination”.

Comment 8: Discussion: Line 230: "Resent" should be "Recent"

Answer 8: Sorry for the typo, it has been corrected.

Reviewer 2.

The article deals with Crohn’s disease interferes with the diagnostic 2 accuracy of an oral rinse active matrix 3
 metalloproteinase-8 point-of-care immunotest

The topic is very important and urgent, and this study introduces a novel and promising approach. The research design is also appropriate.

However, the language of the manuscript is poor, there are few important mistakes, and especially the Introduction and MATERIALS AND METHODS section is unacceptable need rewriting and improve

The materials and methods section has been re-written and the language has been revised by an English expert.

Reviewer 3.

Comment 1) The paper is in need of editing for grammatical irregularities, but also suffers from a lack of detail and clarity. The immunotest was designed to detect the active form on MMP-8, but no information is given as to if or how well it detects other forms, or what the limits of detection are.  The physiological significance of the different forms is not really explained either.  The legends for the tables should be more informative, and the N numbers for the groups should be given in each case.  Are the apparent differences in sensitivity and specificity given in Table 2 statistically different between Crohn’s and Control?  What do the numbers in Table 4 represent?  The mesenchymal active form in the whole population lists a median of 6.2 but a max of 1.9.  The formatting of Table 4 is messed up.

Answer 1) The manuscript has been revised by English language specialist. The materials and methods section has been added with details and citations. The legends for the tables have been improved and N numbers for the groups have been added. Sensitivity and specificity of the aMMP-8 test have been described by Sorsa et al., Peridontol 2000, 2016;70:142-63 and Sorsa et al., Nature Rev Dis Prim 2017;3:17069. The levels are same as described in the present study. This information has been added to the text in lines 882-887. The Table 4 had been messed up by the manuscript submission system as well there was a typo, we apologize for that. We have modified/corrected the Table 4 in the revised manuscript.  

Comment 2) The study is small, especially when subdivided, and the incidence of periodontal disease in the groups is different (the patients with Crohn’s have more mild periodontal disease, with only 2% moderate and no severe disease, whereas the 17% of controls have moderate or severe disease).  The impact of this on results should be more adequately addressed.

Answer 2) The small study size and its impact on the results are now discussed in lines 871-874. 

Comment 3) An attempt is made to correlate levels of the various forms of MMP-8 in saliva of people who tested positive vs negative by immunotest by immunoblot.  No immunoblots are shown and although the methods section describes densitometic analysis, that isn’t in the legend and there is no information as to what the numbers in the table represent or how many samples are in each group.  Immunoblots are at best only semi-quantitative, so it unclear what significance this data can have.  

Answer 3) Representative immunoblot figure and legend has now been amended as Figure 1. In the method section, details of its densitometric quantification with established citations are provided in lines 655-659 in the revised version of the manuscript.

Round 2

Reviewer 1 Report

I again read the manuscript by Rautava et al. The authors have now addressed some of my questions to some extent. However, several new questions/concerns arise which should be clarified;

comment on previous question 1: This first line still remains unclear. This does not require the addition of more words to the abstract, but a better phrasing of this line. For example: “The diagnostic accuracy of point-of-care (PoC) applications may be compromised in individuals with additional inflammatory conditions”. The same is true for my second comment. For example, the last line of the abstract could be: “…which may be related to lower MMP-8 levels or undetectable complexes.” It was very dificult to see all the changes in the new version of the manscript since it was a PDF file and some comments were overlapping with the tables.  The western-blot images are very blurry, which makes me wonder how acurate these bands are? How sure are you about proMMP-8/activated MMP-8? activation of proMMPs can also occur in a sequential manner (e.g. the propeptide is cleaved off in 2 steps, resulting in several bands which are lower than the full proMMP). Is there a way to be really sure?  Were the numbers in table 4 altered? Why?  Are the patients with CD taking any anti-inflammatory medication at the time of the test? What would the effect be on this test?

Reviewer 3 Report

Please see the file in attachment.
